# High Gain Compact UWB Antenna for Ground Penetrating Radar Detection and Soil Inspection

**DOI:** 10.3390/s22145183

**Published:** 2022-07-11

**Authors:** Tale Saeidi, Adam R. H. Alhawari, Abdulkarem H. M. Almawgani, Turki Alsuwian, Muhammad Ali Imran, Qammer Abbasi

**Affiliations:** 1The Microwave Antenna, Device and Systems (MADs) Laboratory, 413 LG Reserach Bldg., 77 Cheongam-ro, Pohang-si 37673, Gyeongsangbuk-do, Korea; talecommunication@gmail.com; 2Electrical and Electronics Engineering Department, Bahçeşehir University, Istanbul 34353, Turkey; 3Electrical Engineering Department, College of Engineering, Najran University, Najran 66462, Saudi Arabia; ahalmawgani@nu.edu.sa (A.H.M.A.); tmalsuwian@nu.edu.sa (T.A.); 4Communications Sensing and Imaging Group, James Watt School of Engineering, University of Glasgow, Glasgow G12 8QQ, UK; muhammad.imran@glasgow.ac.uk (M.A.I.); qammer.abbasi@glasgow.ac.uk (Q.A.)

**Keywords:** ground penetrating radar, CPW, ultra-wideband, soil inspection, image reconstruction

## Abstract

An ultrawide bandwidth (UWB) antenna for ground-penetrating radar (GPR) applications is designed to check soil moisture and provide good-quality images of metallic targets hidden in the soil. GPR is a promising technology for detecting and identifying buried objects, such as landmines, and investigating soil in terms of moisture content and contamination. A paddle-shaped microstrip antenna is created by cutting a rectangular patch at one of its diametrical edges fed by the coplanar waveguide technique. The antenna is loaded by stubs, shorting pins, and a split-ring resonator (SRR) metamaterial structure to increase the antenna’s gain and enhance the bandwidth (BW) towards both the lower and higher end of the working BW. The antenna’s performance in soil inspection is studied in terms of the operating frequency range, different types of soil, different distances (e.g., 50 cm) between the antenna arrays and soil, S-parameters, and gain. Following this, the antenna’s ability to find a metallic target in the soil is tested, considering different array numbers, multi-targets, and locations. The antenna is designed on a thin layer of economic polytetrafluoroethylene (PTFE) substrate with dimensions 50 × 39 × 0.508 mm^3^ and works in the frequency range 1.9–9.2 GHz. In addition, two more resonances at 0.9 and 1.8 GHz are also achieved; hence, the antenna works for more than two application bands, such as the ISM- and L-bands. The measurement results validated excellent agreement with the simulated results. Furthermore, the recommended antenna offering a high gain of about 10.8 dBi and maximum efficiency above 97% proved able to discriminate between hidden objects and even recognize their shapes. Moreover, the reconstructed images show that the antenna can detect an object in the soil at any location.

## 1. Introduction

UWB antennas are the most favored transceivers for several noninvasive applications such as microwave (MW) imaging and finding objects [1,2]. Due to various benefits, the microstrip antennas were miniaturized and are cheap compared to other antennas. In this era, the microstrip antenna has the advantages of small size, high gain, and low cost for exemplary performance in some applications [3,4,5]. Any ground-penetrating radar (GPR) system works by sending an electromagnetic (EM) wave to the ground and towards the hidden objects. Subsequently, the reflections are received by the receiver antennas [6,7]. A GPR system can also detect any nonhomogeneous metal and dielectric object surrounded by soil. Four kinds of GPR system exist as quasi-monostatic radar, monostatic radar, bistatic radar, and multistatic radar [8]. The GPR structures can be categorized as time-domain (impulse radars) and continuous-wave (CW) radars. Furthermore, the GPR techniques typically perform at central frequencies below 1 GHz, and they need a very large BW for a proper resolution and echo. Therefore, an impulse broadband system is required; however, this shows some limitations [9].

As mentioned above, ‘GPR’ is a name given to a device that utilizes EM waves and scattered signals to localize a target when it is buried or hidden in a multilayer environment [10]. The antennas, as transmission and receiving devices, are the most critical part of any GPR system. They can be arranged in three states: monostatic, bistatic, and multistatic. One antenna is implemented for sending and receiving in the monostatic state. However, in bistatic and multistatic states, more than one antenna is used for sending and receiving [11]. In addition, the GPR systems can be categorized into two groups, depending on how they are used: air-coupled and ground-coupled arrangements. The first system utilizes antennas 40–50 cm ground; the latter 5–10 cm [12].

A system having a BW of more than 20% or wider than 0.5 GHz is called ultrawideband [13]. The most common requirements for UWB GPR in low-frequency operation are deep penetration, higher gain to enhance the range resolution, and a constant radiation pattern. Due to these characteristics, UWB GPRs are employed various military and civilian applications. For example, UWB radars, like the UWB GPR using the same principles, have been used to detect mines and hidden humans [14]. These utilize the frequency range above 100 MHz with a 100% bandwidth rate. When a higher frequency range is used, a higher resolution is achieved. However, it cannot propagate too far, and detecting objects hidden too deeply becomes too difficult. A lower frequency results in a lower resolution but higher penetration depth [9,15].

Several techniques exist to improve the design parameters of antennas by applying various feeding techniques. Two standard techniques to improve the antenna’s characteristics are the electromagnetic band gap (EBG), metamaterial (MTM) structures, and defected ground structure (DGS). The EBG structures are used due to their simple design and shape, similar to somewhat rectangular rings, and are etched on both the ground and resonator. MTM structures are also used widely to improve antenna characteristics like enhancing gain and widening the BW [5]. In addition, they are also utilized to suppress the higher-order mode harmonics, mutual coupling between adjacent elements, and cross-polarization for improving the radiation characteristics of the proposed antenna.

Several types of antennas have been used for UWB GPR, such as the planar antenna [16], horn antenna [17], bowtie antenna [18], and Vivaldi antenna [19]. They offered satisfactory performances; however, their bulky and complicated structures limit them from being used for GPR applications. For instance, a bowtie-slot antenna with a BW of 0.4 GHz–1.5 GHz and large dimensions of 50 cm × 22 cm has been designed for GPR applications [20]. A Vivaldi and bowtie antenna with complicated structures obtained 5.8 dB and 3.3 dB gain at 1 GHz with dimensions of 405 × 12 × 318 mm^3^ 12 × 532 × 600 mm^3^, respectively [21]. Thus, ultrawideband antennas with high gain are highly desired in GPR systems. On the other hand, several antennas were designed for UWB GPR but showed drawbacks, such as low gain or large volume, which constrained them from being applied for GPR applications [22]. Therefore, a UWB antenna with militarized dimensions, wide BW, high gain, and stable radiation pattern operating in a lower frequency range is demanded to resolve the antenna limitations for UWB GPR applications.

A bowtie-like patch antenna is fed using the coplanar waveguide (CPW) technique with defected ground through a 50 Ω transmission line (TL). The antenna has been developed to be applicable for UWB GPR applications such as soil inspection, having a low profile as well as being cheap, high gain, and wide BW, especially at the lower band of the accepted BW for UWB antennas based on the Federal Communications Commission (FCC). First, the introduction about the UWB antennas and GPR applications is given. Subsequently, the antenna design configuration and optimization are presented in Section 2. Afterward, the results and discussion are shown in Section 3. Finally, conclusions are drawn in Section 4.

## 2. Antenna Configuration

Several methods, such as loading by stubs, shorting pins, and periodic SRR, are considered to achieve and improve the characteristics of the proposed antenna for GPR application. The proposed antenna’s structure and dimensions are indicated in Figure 1 and Table 1, respectively. Figure 1c is the fabricated antenna. After fabrication, the antenna is measured using VNA, as shown in Figure 1d. The antenna is measured in the air first, and then the media is changed into the soil. The antenna is connected to the first terminal of the VNA, and another antenna is connected to terminal two to carry out the measurement. After securing the antennas to the VNA’s terminals through cables, the VNA should be calibrated using the calibration kits based on the diameter of the antenna’s SMA ports (2.4 mm or 3.5 mm). After assigning the frequency band and the calibration, the antenna’s S-parameters (reflection and transmission coefficients) are measured. Another part of the measurement is the measurement of the radiation pattern of the antenna (Figure 2). The proposed antenna is fixed on the rod at the center, and a reference horn antenna (connecting to the power meter) is fixed on the rotating rod to perform the radiation pattern measurement. The motion controller, which rotates the rotating rod, rotates 3 degrees in each step, then stops at each stage of 3 degrees, makes a recording, and continues. The power recorded by the meter is connected to MATLAB code in the PC, and then the radiation pattern is drawn. Finally, the antenna is moved to the elevation plane, and the measurement process starts again.

The antenna has small dimensions of 50 × 39.5 × 0.508 mm^3^. It is designed on a marketable cheap PTFE substrate with a thickness of 0.508 mm, ε_r_ = 2.1, and tanδ = 0.001 [23,24]. Figure 3 depicts the design evolution of the patch and the ground. First, a conventional rectangular patch fed by CPW is designed at the center frequency of 5 GHz to meet UWB standards according to the FCC. However, the working BW is not wide, and an undesired coupling capacitance occurs at the edge of the substrate where it separates the patch from the ground (Figure 3e shows a better understanding of the undesired coupling capacitances (*C_p_*, *C_c_*) and the unwanted surface waves). This coupling capacitance degrades the antenna’s efficiency. Therefore, the ground is cut from both sides to reduce this effect. Another factor that affects the antenna’s performance is the surface waves at the junction of the transmission line and the patches. Thus, the diagonal cut can help decrease that effect. More explanation about the design steps and loading of the antenna to improve the antenna’s performance is presented in the following paragraphs. It should be mentioned that the patch length has minor effects on the resonance but improves the BW’s widening. On the other hand, its length affects the resonance frequency. Hence, loading the antenna later can compensate for the slight probable impact on the BW after cutting the edge.

The proposed antenna comprises two paddle-shaped patches fed through a transmission line using the coplanar waveguide (CPW) technique. The paddle shapes created by two mirrored rectangles were cut 55° from the edge. They are spaced by 7 mm to reduce the surface waves and prevent degrading radiation efficiency and gain (Figure 3e shows how the coupling capacitances exist around the conventional antenna). The standard rectangular patch dimensions are reduced when it is changed into the paddle shape compared to the conventional arrangement without hindering the design’s simplicity. The coplanar ground plane also does not cover the entire width of the substrate. This is to enhance BW at a higher frequency and results in a thorough broadening of the antenna-impedance bandwidth. This is due to the decrement of the unwanted fringing fields and coupling at the edge of the substrate (caused by the coupling capacitance in Figure 3e) when a whole ground is employed. It should be mentioned that the space between the paddles (S′ in Figure 1) should not be optimized to enhance the surface waves, which degrades the radiation efficiency and gain of the antenna.

Furthermore, if the length of the right paddle towards the *Y*-axis is increased, the whole BW will be shifted to the lower band. This is also because of the inductance increment in that region. However, two near stopbands will be created if they exceed 4.5 mm (i.e., the impedance bandwidth level is less than −10 dB). On the other hand, the left paddle is cut more towards the *Y*-axis until it reaches the right paddle for the same reason mentioned for the right paddle (it reduces the capacitance, which reduces the resonance frequency and shifts the band toward a lower BW). When the lower and higher end of the working BW of the antenna is obtained (based on FCC for UWB applications), the antenna is loaded by eight chamfered-edge cubes based on the antenna’s surface current distribution (SCD) around the paddles, transmission line (TL), and the ground (GND). This loading improves the impedance BW at the lower and higher end of the working BW since it creates an additional current flow at the back, excited through the coupling distance of the ground at the back. However, it makes small stopbands around 4.5–5.5 GHz and 7–8 GHz (it pulls up the reflection-coefficient level to nearly −10 dB, because the fields at the fields are perturbed due to the newly created flow as mentioned above). Afterward, the antenna is loaded with three stubs rotated to the same degree as the left patch. These stubs are conductively and capacitively-loaded with the same space among them. In addition, due to the capacitance increase, the resonant frequency and the BW are slightly shifted to the higher band. Therefore, these stubs’ space and length are optimized to obtain resonances at 1.8 GHz and 0.9 GHz. It should be mentioned that the impedance matching around 0.9 GHz is around −8 dB at this design stage. Thus, it should be improved to reach −10 dB, where it is acceptable. The antenna is loaded using four shorting pins connecting these stubs to the chamfered cubes at the back. This loading reduces the possible undesired surface wave in that antenna area. Following this, the antenna is loaded with the SRR periodic MTM structure to enhance the gain and radiation efficiency. After MTM loading, six more pins connect the ground to the MTM structure to improve the impedance BW of the antenna and convert all the stopbands into passbands. The full explanation of each design stage is presented later.

### 2.1. Parametric Study of the Proposed Antenna

This section presents the parametric study of the proposed antenna. For all these parameters, the actual value for each parameter is calculated first using the equation of the microstrip patch antenna and the CPW-feeding technique. They are then optimized to obtain the best outcome for every parameter during the design process. The antenna’s most influential parameters affect the band’s higher and lower end. The transmission-line-dimension (W_f_, L_f_) variations in frequency are presented in Figure 4a. These show that the antenna has the widest BW at around 28 mm. It starts from 2.15 GHz to 10 GHz.

Furthermore, the BW is reduced with a reduction in the length of the feed line. Figure 4b indicates the antenna’s reflection-coefficient result when the feed line’s width is varied in frequency. When the feed-line width is around 1 mm, the optimum BW is obtained. The impedance BW of the antenna is degraded, especially at the middle of the working BW when the width exceeds 1 mm (the references regarding the theory of the impacts of feed-line width and length are [25,26,27].

After the effects of the feed-line dimensions on the reflection-coefficient results were studied, the impacts of the ground dimensions were examined. The ground length impact is almost identical to the feed-line length. Thus, it is not investigated here; only the width of each ground part is presented here, as W_g1_ and W_g2_. Figure 5a shows the reflection-coefficient results of the antenna, considering various widths W_g1_. It shows that the working BW is broad when this width is around 17.5 mm. When it is increased to 19 mm and reaches the substrate edge, the impedance BW is degraded at higher frequencies due to the coupling effects explained before. The reflection-coefficient results of the second width of the ground, W_g2,_ are indicated in Figure 5b. An increase in this width enhances the BW until it reaches 11 mm. Exceeding 11 mm creates a stopband in the middle of the working BW. The other parameters of the antenna design were also optimized to achieve the best results (The optimization is performed in CST using the PSO algorithm after the actual values obtained for each parameter). For instance, the length of the stubs (L_4_) should not exceed 17.85 mm (so as not to increase the undesired capacitive coupling) and less than 4 mm (this would affect the reflection-coefficient level of the lower bands like 0.9 GHz and 1.8 GHz). The width of the stubs (L_9_) affects the impedance bandwidth of the antenna at the lower end of the working BW and two lower resonances. The space between each stub (S_5_) also needs some attention as it should not be more than 4.7 mm and less than 4. If it is less than 4 mm, this increases the surface waves around that area.

When the parametric study of the antenna without any loadings is finished, the antenna is loaded with chamfered cubes at the back, pins, stubs, and the periodic MTM structure. Figure 6 illustrates the reflection-coefficient results at each loading stage in terms of frequency. As can be noticed in Figure 6, the working BW is enhanced for both ends of the BW after loading with the chamfered cubes. An impedance bandwidth of around 4.5–5.5 GHz and 7–8 GHz is narrowly accepted because it is around −10 dB. Therefore, the whole BW is enhanced at both the lower and higher end of the BW. Afterward, the antenna is loaded with three stubs with a similar space between them. These stubs are conductively and capacitively loaded to create two more bands at the lower band and to have a resonance at the L-band of 1.8 GHz. Besides this, another resonance was created at 0.9 GHz as an industrial, scientific, and medical (ISM) band. However, this band is not working correctly since it is around −8 dB. Hence, it is loaded with four pins. Pin loading creates more poles (resonances). Therefore, it slightly increases the BW (utilizing shorting pins is a technique that can produce more poles or resonances, enhancing the antenna gain and radiation efficiency, increasing the BW and surface-wave suppression). Due to the inductive shunt effect of these shorting pins, the dominant mode’s resonant frequency is tuned up to enhance the radiation gain of a single patch antenna [28,29,30]. This also improves the level of reflection coefficient at two resonances at the lower band (the pins acting like a parallel resistor and inductor in series with a capacitor). Following this, the antenna is loaded with the periodic structure and six pins connecting it to the ground. The reflection-coefficient level and the BW improves after using the MTM structure attached to the ground using the pins. Moreover, it enhances the radiation efficiency and gain of the antenna, which is depicted later in the Section 3.

When the surface current starts flowing from the port and then the transmission (feed) line, with any changes in the shape of the antenna, such as the shape of the patch and/or the stubs, the current is altered, and so too are the fields around the antenna. Therefore, for a better understanding of each stage in the design and the stopbands/pass bands created during the design procedure, the surface current distributions of the antenna should be investigated at these frequencies, as indicated in Figure 7. In addition, when these stopbands and passbands existing in the working BW are well understood, removing the stopband and converting them to passbands will be easier, which helps to broaden the BW. For instance, the surface current density (SCD) at the lower and higher end of the BW at 2.35 GHz and 8 GHz for the antenna without loads show that the SCD is stronger around the TL and patch (Figure 7a). It can be seen that the magnetic field strength is low, as the light green color indicates 0 to 3 A/m. The current distribution has a shallow flow and magnitude at the input port, and the current distribution reduces as it approaches the patch. The signal from the input port is partially lost as it travels to the patch. The SCD for the chamfered cubes also shows strong density around these cubes at 2.1 GHz and 9 GHz, where both ends of the BW exceed what was obtained for the previous design stage. Most likely, other parts of Figure 7 depict the strong SCD at the poles in the working BW at each design stage. For instance, the surface current density around the stubs at 0.9 GHz is higher than the current density at 1.6 GHz. It can be deduced that the electromagnetic wave of a specific frequency is excited by the stubs and the patch resulting in resonance at 0.9 GHz as the color is nearly red and the magnetic field around the stubs at that frequency is increased to 8 A/m. Figure 7d shows the surface current distribution around the MTM structure at 1.9 GHz and 9.2 GHz. It can be observed that the current is no longer limited between the radiation patch and feed lines. It also distributes around the SRR, demonstrating that the SRR can act as a resonator to generate new resonance at 1.9 GHz. Obviously, after adding the SRR, the current distribution is changed.

Moreover, the current shocks back and forth in the SRR, radiating a specific frequency of electromagnetic wave. At a frequency of 1.9 GHz, the surface current mainly distributes around the SRR. Thus, the newly generated resonance is primarily affected by the parameters of the SRR.

### 2.2. Antenna’s Performance Improvement Utilizing Periodic SRR Structures

The functioning guideline of circular split-ring resonators (SRR) can be effectively perceived with the assistance of their comparable circuit models that display the complete idea of the SRRs. Assuming an outside field is applied to the SRR along the *Z*-axis, electromotive power (EMF) appears nearby. This EMF pairs two concentrical metallic rings with current prompted in them, which passes from one ring to the next through the capacitance shaped between the internal spacing of the rings. The equivalent circuit of the SRR presented is a parallel LC resonator. The current going from one ring to another demonstrates the presence of whole inductance (LT). The capacitances delivered are shaped around the SRR structure, including the two halves and the rings. It also consists of the capacitances associated with gaps in the split rings. All the equations for the total capacitance C_T_ and inductance L_T_ are presented in [31]. The resonating frequencies (f_r_) are 5.15 GHz and 5.94 GHz for the bigger and smaller SRR, respectively [31,32,33].

For further verification of the SRR operation, it was simulated in a computer-simulation-technology microwave studio (CST MWS). First, the SRR structure was positioned near the TL. Subsequently, the S-parameters as reflection and transmission coefficients are plotted in Figure 8b for both the SRRs. The bigger SRR displays a resonance around 4.5–5 GHz, whereas the smaller one resonates around 7–7.5 GHz. Moreover, the negative index of the MTM structure, like the permittivity and the permeability, also shows that the MTM structure works nicely to improve the gain at these frequencies (Figure 8a).

## 3. Results and Discussion

This section shows the characteristic radiation results of the proposed antenna in free space, different kinds of soil, different moisture contents, and when a metallic target is hidden in soil (like a mine in minefields). The soils chosen were sandy and loamy for both wet and dry conditions. They were bought from the shop to fill a container with different kinds of soil and various moisture content levels. The moisture content of the soil was measured first, then calculated by weighing wet soil sampled from the field, drying it in an oven, and then weighing the dry soil. Thus, the gravimetric water content equals the wet soil mass minus the dry soil mass divided by the dry soil mass. In other words, the mass of the water is divided by the mass of the soil.

The reflection-coefficient results of the antenna in free space (no soil) and sandy soil (dry and wet) were investigated for both simulation and measurement. A good agreement occurred between the simulation and measured results. However, a minor discrepancy existed in the BW. All the resonances and the working BW were achieved. When the medium of the simulation was changed to the sandy soil, under dry and wet conditions, the impedance BW of the antenna was disturbed as compared to the results in free space. This is because of the higher dielectric constant of the sandy soil, especially the wet one, compared to the dry soil and air (free space). Figure 9 illustrates the measurement setup of the fabricated antenna, the antenna arrays, and the test bed, including soil. After evaluating all the investigations with different types of soil using the simulation results, a metallic target was hidden at the center of the container (made of foam) at a depth of 5 cm in loamy soil for further investigations. The reflection-coefficient results of the antenna in different types of soil are presented in Figure 10. Few variations occurred using different soil types as a test bed. Furthermore, a reasonable agreement was obtained between the simulated and experimental results.

In addition to the reflection-coefficient results, the antenna’s radiation pattern was also measured for electric (E) and magnetic (H) fields in Figure 11 at different frequencies of 0.9 GHz, 1.8 GHz, 1.9 GHz, and 9.2 GHz. The electric field showed the main lobe mostly around 0 degrees, and only at 9.2 GHz did it slightly shift to 30 degrees. In addition, the magnetic field also followed the same tendency as the electric field. However, a minor discrepancy occurred between the simulated and measured radiation patterns.

Table 2 and Table 3 show the comparison tables. Table 2 demonstrates the comparison performances between several similar works. All these works designed a UWB antenna for GPR applications. They considered both low- and high-frequency bands. The proposed antenna showed better BW and gain performance with lower dimensions. On the other hand, Table 3 compares our antenna with some antennas designed for GPR application, and they were considered in a medium, such as soils, with different thicknesses. The proposed antenna offered better performance in comparison with them too.

An antenna with high gain might not be essential for UWB and GPR applications but is an important factor for an antenna because antenna gain is more commonly quoted than directivity in an antenna’s specification sheet, because it considers the actual losses that occur. In addition, the antenna gain indicates how strong a signal an antenna can send or receive in a specified direction. Gain is calculated by comparing the measured power transmitted or received by the antenna in a specific direction to the power transmitted or received by a hypothetical ideal antenna in the same situation. Antenna gain is also a measure of the maximum effectiveness with which the antenna can radiate the power delivered by the transmitter towards a target. Besides this, the signal power is vital in the signal processing and imaging of a target here, since the received signals are used to reconstruct the image and are utilized in the algorithm. Due to the importance of gain in antenna design, several works consider high gain while designing an antenna for radar and microwave imaging.

### 3.1. Antenna Performance Investigations in Various Soils

After assessing the antenna’s performance in terms of the impedance BW for both simulation and measurement, the antenna’s capability to work in a medium other than air, such as soil, is evaluated. The first is to check how the antenna works at different distances from the soil. Thus, the antenna’s reflection coefficient was assessed at various distances of 10 cm to 50 cm from dry sandy soil. When the space was increased, more stopbands were created in the working BW. For instance, when the distance was enhanced to 50 cm, the whole BW was reduced at the lower and higher end of the BW. In addition, the reflection coefficient was nearly −10 dB, around 3–4 GHz and 6–8 GHz (Figure 12). Apart from the reflection coefficient, the transmission coefficient is also evaluated at different distances from the dry sandy soil. Figure 12 depicts that when the distance of the antenna is increased, few differences occur in the transmission-coefficient result. However, when it is enhanced to 50 cm, the transmission-coefficient level is raised towards zero.

After the antenna was investigated at different distances in a dry sandy soil, the antenna’s performances were evaluated in various types of soil loamy soil (dry and wet) and sandy soil (dry and wet). The simulation setup on different types of soil is shown in Figure 13. Figure 14 depicts that the moist sandy soil’s reflection-coefficient level is less than other soil types. It is due to the dielectric constant of the wet sandy soil compared to the different types of soil (ε_r_ = 13.8). It can be noticed that when the medium is air or dry loamy soil (ε_r_ = 2.33), the impedance BW of the antenna is better than other types of soil. The same tendency goes for the transmission coefficient. Figure 14 reveals that the transmission-coefficient level is also decreased to negative for both wet loamy and wet sandy soil. It indicates a higher level of transmission line for the free space, dry loamy soil, and dry sandy soil.

The other parameter that should be considered is the antenna’s received signal that is sent from one antenna array towards the working medium and is received by different arrays of the antenna. The signal that enters a new medium can show how much the signal is changed in amplitude and how it has shifted due to the delay after passing that environment. Figure 15 shows the signal received after passing through the free space and different soil types. It illustrates that the received signal has the highest amplitude after passing through air. On the other hand, the signal’s amplitude is reduced after passing through the wet loamy and sandy soil. Following this, these received signals were utilized to find a hidden metallic target in the soil.

After investigating the antenna’s performance in different types of soil, it should be examined if the antenna can work well when the soil has two layers with two different dielectric constants. Therefore, two layers of soil, loamy and sandy, were considered, with dielectric constants of 2.33 and 2.8, respectively. Figure 16 indicates the S-parameter results of the antenna when two layers of soil are presented at the front of the antenna arrays at a distance of 10 cm. It shows that neither reflection nor transmission coefficient is altered dramatically; only a slight shift in BW exists. Therefore, the antenna can work even when more than one layer of soil exists at the front of the antenna arrays.

### 3.2. Antenna Investigation in Different Levels of Moisture Content of Soil

The reflection and transmission coefficient results of antenna arrays were investigated considering different moisture content (ε_r_ = 2.33 − 18.9) for two other potential hydrogens (pH) of soil, 4.7 and 7.4, to control the moisture content of the soil under testing using the proposed antenna [44]. The dielectric constant of soil with an inherent moisture content of 4.7 pH and all other types of moisture content, such as the 28% moisture content of 7.4 pH soil, are changed within the range 2.33 to 18.9. Figure 17 depicts that an increase in moisture content enhances the reflection-coefficient level towards the positive. This means that an increase in moisture content degrades the working BW. Therefore, more waves and signals will be absorbed by the soil’s moisture content. The investigation here was performed when the dielectric permittivity of the soil was increased up to 18.9 for 28% of moisture content and 7.4 of pH. Higher percentages of the moisture content in the soil will degrade the S-parameter level. However, the proposed algorithm can work well, having a narrower bandwidth (BW) due to having a time-reversal nature, because time-reversal algorithms remove the effects of the background at the beginning of the algorithm. Therefore, they can effectively remove more clutter and artifacts in the imaging environment.

The S-parameter results of the antenna-like reflection and transmission coefficient were taken considering different assumptions, such as different spaces between the arrays and the soil, different kinds of soil, there being more than two layers, and different moisture contents, to be sure that the proposed antenna works appropriately in an environment like soil. Furthermore, this can be a way to control the soil’s conditions in terms of humidity, soil contamination due to damage to an oil pipe in an oil field, or even finding a hidden target like a mine in a minefield. For instance, the S-parameter results, especially the transmission coefficient, are used primarily to find the target, since they are related to the signals received by other arrays (for instance, when A_1_ sends and A_2-4_ receive in Figure 13b).

### 3.3. Antenna Capability for Detection of a Hidden Target in Soil

After showing that the antenna can control the moisture content of the soil under examination, another ability of the antenna, i.e., to detect a hidden object in the soil, was investigated. Different assumptions were used to show that the antenna can detect a target in soil. The simulation and measurement studies utilized a central spherical target, an offset target, and three targets. Afterward, four arrays of the proposed antenna were located 10 cm above the soil with an average moisture content (ε_r_ = 9) and dimensions 300 mm × 400 mm. Before reconstructing the image of the target, the signals received from these arrays (when array 1 sends and the other arrays receive) and the mutual coupling (or isolation depicted by S_n1_) should be evaluated. It should be mentioned that the transmitters and receivers send and receive the signals in a multistatic manner and all the antennas are identical as receivers and transmitters. One transmitter sends, and the other receives, and then the transmitter switches with the other until the fourth array (as is usual for a multistatic). However, to reconstruct the image of the target in this article, only one transmitter sent, and the others received, thus saving processing time.

Figure 18, Figure 19, Figure 20 and Figure 21 show the transmission coefficient results and received signals of the proposed antenna arrays for different conditions, such as there being three targets, no targets, an offset center target, and a central target. The high isolation of more than 22 dB (S_n1_ < −22 dB) is evident in Figure 18, Figure 19, Figure 20 and Figure 21. In addition, the received signals also demonstrate shifting in the peak of the amplitude at each different array due to the different delay and distance between each array from array 1 (O_21_–O_41_ are the signals received by A_2_–A_4_).

The images of each target for each consideration are reconstructed utilizing the robust time-reversal algorithm presented in Figure 22 [45]. At first, the received signals from each array were extracted from CST and then imported into MATLAB. After calibration, they are delayed and summed. Afterward, they are paired, multiplied, and added again. Finally, the paired multiplied signals are windowed to pick the response of the metallic target.

The EM behavior and modeling of a conventional time-reversal (TR) algorithm is initially utilized. The TR algorithm acts with reciprocal characteristics of the wave equation where the E- and H-field components are propagated backward. It calculates the backscattered signals from the target too. Therefore, an antenna with high performance, such as a wide BW and higher fidelity in the received signals, is required. The TR has been utilized for several applications, such as ultrasound (US) imaging. It presented promising outcomes even when the environment under test is not symmetrical and much clutter exists [46]. Besides this, it faced limitations when the image was reconstructed. For example, some works found that conventional TR consumes too much time to handle, was sensitive toward weak contrast, and offers restricted resolution [47]. Furthermore, when an impulse-transmitting system like a UWB transceiver is used, the correlation between the transmitted and received signals should be high so as not to lose the necessary information [48].

The proposed antenna uses a modulated UWB Gaussian pulse at the resonance frequency. Before simulation initials, all the fields are calculated for conditions with and without targets to remove the cavity’s effects and calibrate the signals for each array. Figure 18, Figure 19, Figure 20 and Figure 21 show the signals received for each array when array A_1_ transmits a signal and the other three receive it (Figure 13). To acquire the signal received from every two antenna arrays, they surrounded the soil (shown in Figure 13) by 200 mm; for the unhealthy soil (polluted or with a metallic target), a spherical metallic target (diameter = 25 mm) is located at the center of the soil as well. Afterward, the transmitter sends the Gaussian pulse, and the receivers receive the signals shown in Figure 18, Figure 19, Figure 20 and Figure 21.

Several works were performed using the planar arrangement of array antennas with various numbers of arrays to detect targets, such as tumors in a breast or hollows in tree trunks [49,50]. The imaging system’s capability is assessed using a planar arrangement of the antenna arrays to detect a hidden target in the soil, such as a mine. The signal processing process was started from the conventional TR to obtain the deviations in target signals. The signals were then reversed in time and backpropagated to focus on the targeted loci. This is to compute and extract the target’s response from the entire field after removing clutter and artifacts. This processing is considered a substantial procedure because the target’s response is subjugated by other scatters from different layers, creating inhomogeneities. This attempts to minimize and reduce the impacts of clutters and artifacts to emphasize the targets’ response which is the most important element of the signal processing. The algorithm diagram is depicted in Figure 22 (A_1_–A_4_ are the calibrated received signals).

The first stage is to remove and subtract the background signals. The background signals contain the effects of the arrays on each other and coupling effects. The second is when the early time contents are removed from the output signals from the last section. These early time signals have a higher amplitude than the targets’ response. This is carried out when all scattered signals are averaged and subtracted from each other. The averaging gives the early time content since the antenna is symmetrically aligned around the soil.

Furthermore, executing time gating later in the signal processing improves the removal processes when the real target and the environment are not symmetrical, and it offers smoother signals [51]. Afterward, the scattered signals from the last section are paired and multiplied. This is undertaken to increase the accuracy in detection and localization. This processing of our scattered signals, A_1_, A_2_, A_3_, and A_4_, can be performed as follows (u1to u4 are mentioned merely as an example to show how pairing multiplication works):A_1_ = E_1_(t), A_2_ = E_2_(t), A_3_ = E_3_(t), and A_4_ = E_4_(t)
X_1_ = A_1_·A_2_, X_2_ = A_1_·A_3_, X_3_ = A_1_·A_4_, X_4_ = A_2_·A_3_, X_5_ = A_2_·A_4_, X_6_ = A_3_·A_4_

The output achieved from the last section of the signal processing is averaged again to further reduce the clutter and artifacts and improve the target detection. Following this, the time of arrival is determined to perform time grating and detect the target’s response. This measures the time between a wave’s touching a sample’s front and back walls as the ‘early’ time and ‘late’ time, respectively. In addition, it provides a helpful window that further reduces clutter reflected from a back wall. Finally, time gating and windowing are applied to window the time obtained for arrival signals by multiplying it with a short Gaussian pulse. This helps detect the smaller targets.

Figure 23, Figure 24 and Figure 25 depict the reconstructed image of the targets when three targets, the offset center target and the central target, are considered (utilizing the simulation results). All the targets are ideally detected. However, only negligible clutter existed in the reconstructed image when three targets were considered. The clutter in the reconstructed images might be due to the effects of the arrays on each other, known as the coupling effects, or background effects like the effects of the soil and the other materials in the soil or other metallic targets in the soil.

After showing the system’s capability to detect a target in soil using the simulation data in Figure 23, Figure 24 and Figure 25, the measured data using the measurement setup shown in Figure 9 are used to reconstruct the image of a target hidden in the soil at a depth of 10 cm. Figure 26 indicates the reconstructed image of a spherical target at the center, three targets, and an off-center target using the measured signals. The measured transmission coefficient (S_21_) is extracted from the vector network analyzer (VNA) to reconstruct the image using the measured signals and then converted into the time domain using the inverse fast Fourier transformer (IFFT). The reconstructed image shown in Figure 26 depicts that the targets are ideally detected using four arrays of the proposed antenna. However, more negligible clutter are in the reconstructed image, and the detected targets are shifted slightly from their real location. This negligible clutter and changes in the actual location of the targets occurred due to background effects, such as the effect of the soil itself, the moisture content, and the possibly different, unevenly distributed materials in the soil. The robustness of the antenna on a false target can be shown by how much the antenna can be accurate in the detection of the real target and some clutter. Can the target be differentiated from the other objects in the testing medium, as when we have three targets, for example? It can be concluded that the antenna can detect false targets and differentiate them from other targets or objects; Figure 26 proves this. The clutters that were detected in Figure 26a are due to the uneven distribution of the soil in the test bed. This uneven distribution can be similar to the situation in which another particle exists in the soil. Another proof of the robustness of the antenna is shown in Figure 26b, when all three targets were detected. However, some clutter was also detected. The targets are nonetheless perfectly recognizable since their dynamic range is nearly one (yellow). Since the antenna could reconstruct the image of a spherical target in soil using both simulation and measurement information, it can be concluded that the proposed antenna is an excellent candidate for soil inspection and finding a metallic target, such as a mine hidden in the soil. In addition, the smallest target that the antenna can detect in any location within a certain distance of the antenna is around 10 mm in diameter. This can be calculated considering the spatial resolution, and, apart from the spatial resolution, the range resolution (10 mm for 10 cm distance between the antenna and the target) and cross-range resolution (11.28 mm). The full explanation and formulas are presented in [30,52,53,54]. 

## 4. Conclusions

The GPR statement is given to a system that can find a hidden or buried object using the scattering from the electromagnetic waves sent by antennas. Most of these systems have utilized a low-frequency band (for better penetration in soil, for example), which renders the sending and receiving systems bulky (antennas). Therefore, a low-profile antenna with high performance is required. A novel ultrawideband paddle-shaped antenna incorporated with periodic metamaterial array structures is proposed for landmine detection using a ground-penetrating radar (GPR) system. The proposed antenna is designed at a center frequency of 5 GHz on a PTFE substrate ($\varepsilon_{r}=2.1,\;\;tan\delta =0.001$) beginning with a conventional rectangular patch fed using the CPW technique. This is to meet the UWB antenna’s required working band based on the FCC. The ground was then cut asymmetrically to reduce the capacitive coupling (to enhance BW at a higher frequency and thoroughly broaden the antenna impedance bandwidth). Afterward, the patch was cut, making a paddle shape to reduce the surface wave (another factor degrading the antenna’s performance). Eight chamfered-edge cubes then loaded the antenna following the antenna’s SCD around the paddles, TL, and the GND (this improves the impedance BW at the lower and higher end of the working BW). Following this, the antenna was loaded with three stubs (to convert the slight stopbands to passbands after adding the cubes, to improve the reflection coefficient level of two more poles at 0.9 GHz and 1.8 GHz created before) and four shorting pins (to reduce the possible surface waves around the stubs and patch). Last but not least, an SRR periodic MTM structure and six more pins were utilized to enhance the radiation efficiency and gain, along with improving the level of the reflection coefficient. As a result, it offers enhanced gain and directivity.

The proposed antenna offers an operational bandwidth from 1.9 to 9.1 GHz. The average directivity reaches 11.2 dBi, while the gain and radiation efficiencies are 10.8 dBi and 97%, respectively. The size is small due to the antenna loading with stubs, and even resonates at two more resonances, at lower frequency bands of 0.9 GHz and 1.8 GHz for ISM. The antenna performance was simulated and then compared with the measurement. Good agreement exists between numerical and experimental results. The antenna response was studied for ground-coupling GPR applications like finding a hidden metallic targets, including various considerations, such as different types of soil, different levels of moisture content of the soil, soil with multiple layers, and different thicknesses of soil (the space between the antennas and the soil level, target, and beneath the level of the soil in the container). Stable and linear transfer-function response and flat group-delay response was obtained in the antenna passband, which confirms the low dispersive nature of the proposed UWB antenna and thus ensures its operational capability as a GPR antenna. Moreover, the reconstructed GPR image collected with the proposed antenna from the simulated and experimental setup with a sandbox, metallic target, and the moist sandy soil layer, shows that the proposed antenna is a reliable candidate for GPR applications.

## Figures and Tables

**Figure 1 sensors-22-05183-f001:**
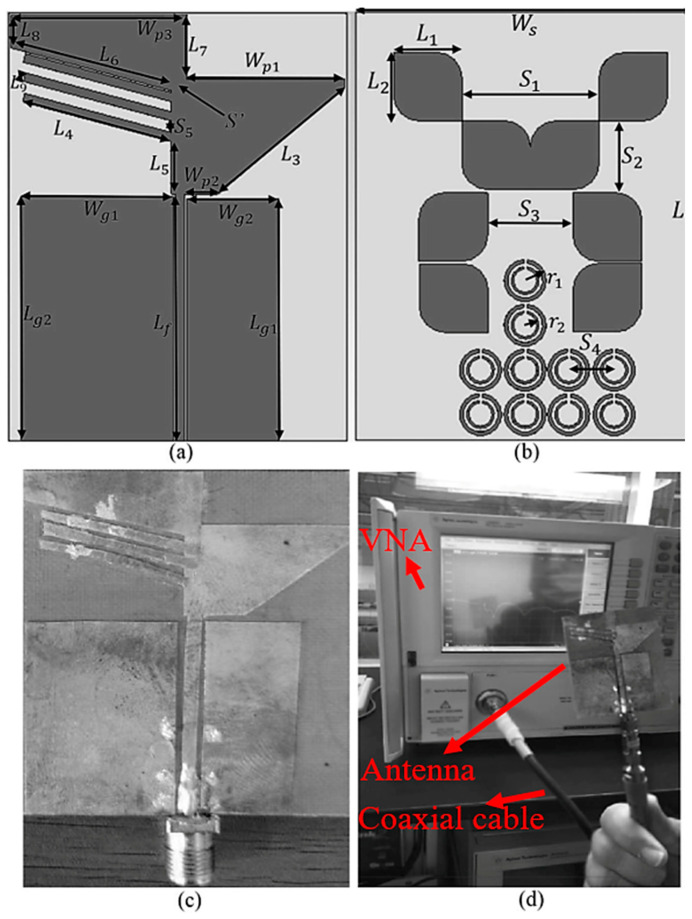
The simulated prototype (**a**,**b**), fabricated prototype of the proposed antenna (**c**), and the measurement setup of the antenna in the air (**d**).

**Figure 2 sensors-22-05183-f002:**
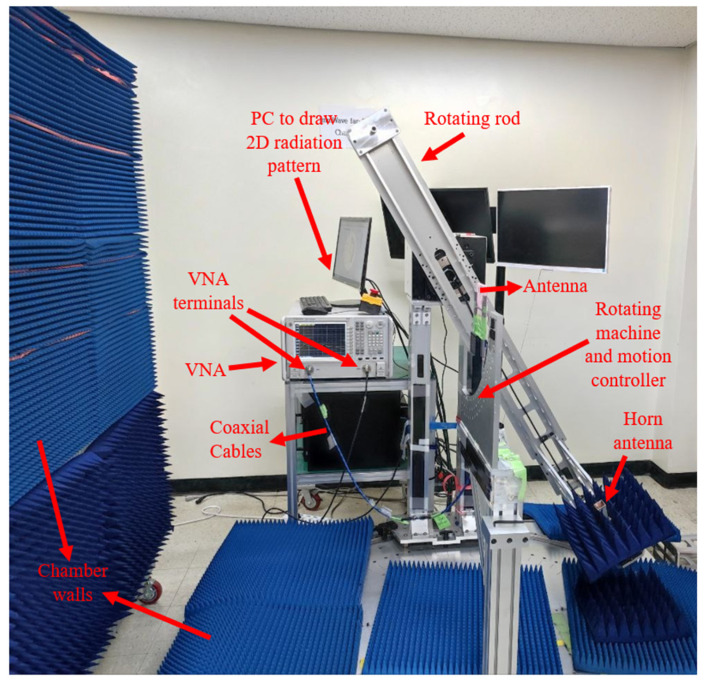
The measurement setup for the radiation pattern of the antenna (the antenna is fixed on the rod covered by foam (the pink structure in Figure 2), not touching the rod).

**Figure 3 sensors-22-05183-f003:**
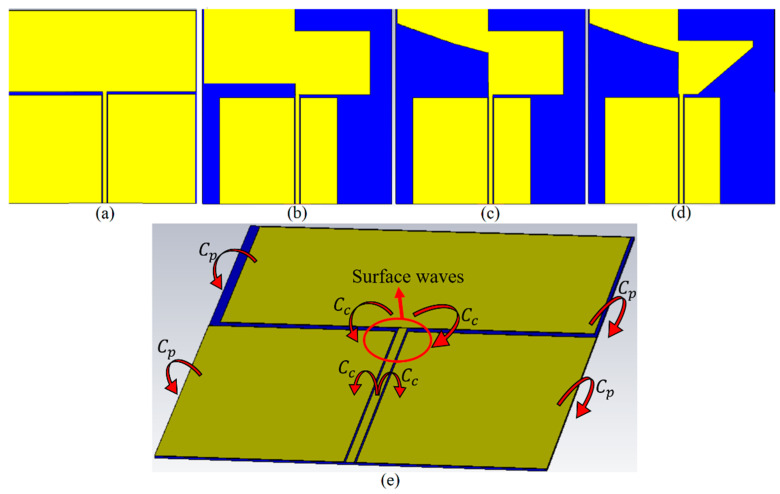
The simulated patch and CPW ground design evolution prototype (**a**) the conventional patch, (**b**) reduced ground from both sides and the right half patch, (**c**) diagonal cutting of the left half patch, (**d**) diagonal cutting of the right half patch, and (**e**) the perspective view of the conventional antenna showing the coupling capacitances and the surface wave.

**Figure 4 sensors-22-05183-f004:**
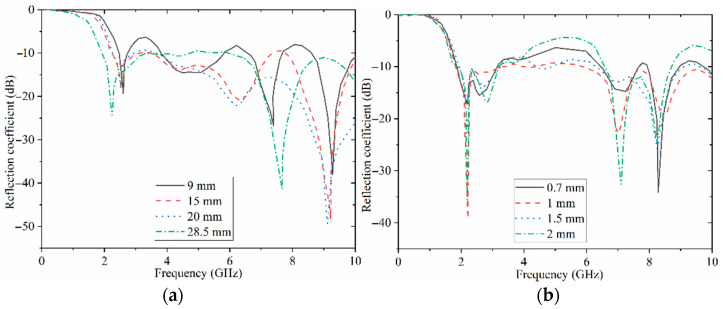
Simulated reflection-coefficient results in terms of transmission line dimensions (**a**) L_f_ and (**b**) W_f_.

**Figure 5 sensors-22-05183-f005:**
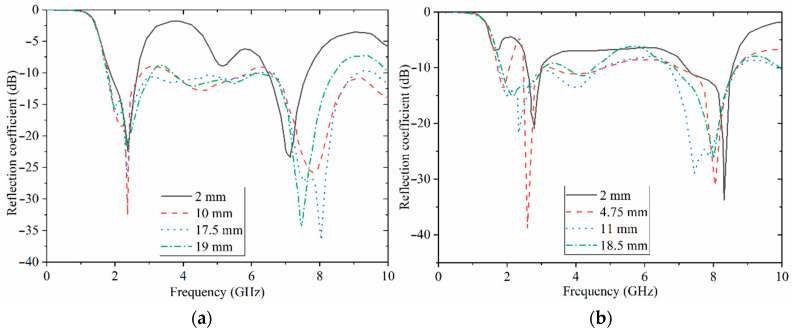
Simulated reflection-coefficient results in terms of ground width (**a**) W_g1_ and (**b**) W_g2_.

**Figure 6 sensors-22-05183-f006:**
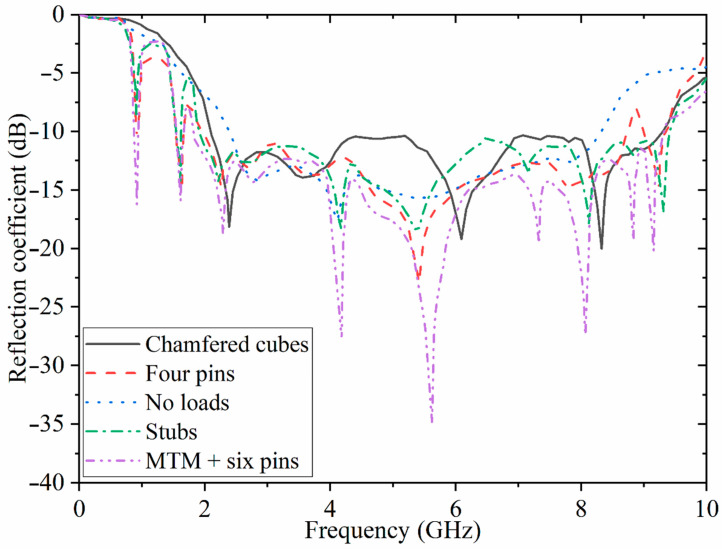
Simulated reflection-coefficient result for each stage of the design.

**Figure 7 sensors-22-05183-f007:**
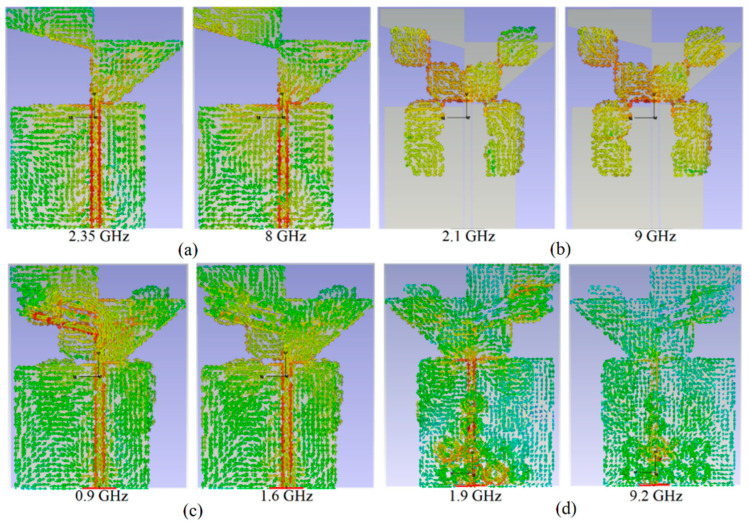
Simulated surface current distribution at different frequencies (**a**) the antenna without loadings, (**b**) after loading with the chamfered cubes, (**c**) after adding stubs and pins, and (**d**) after adding the MTM structure.

**Figure 8 sensors-22-05183-f008:**
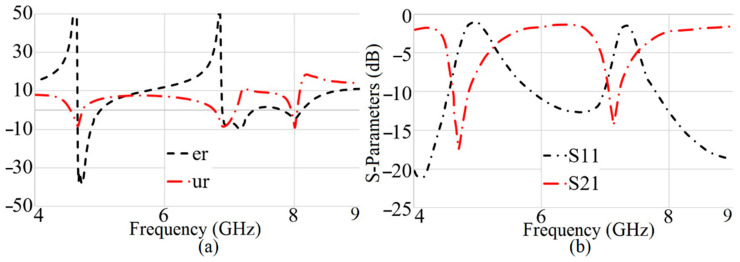
Simulated results of the metamaterial structure’s (**a**) extracted permittivity (e_r_ (F/m)) and permeability (ur (H/m)), and (**b**) S_11_: reflection coefficient, S_21_: transmission coefficient.

**Figure 9 sensors-22-05183-f009:**
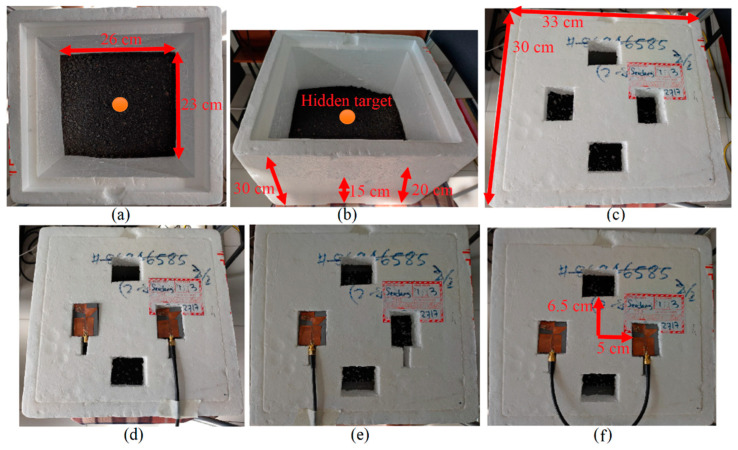
The measurement setup: (**a**–**c**) the test bed container made of foam, and (**d**–**f**) the antenna arrays located on the container and coaxial cable.

**Figure 10 sensors-22-05183-f010:**
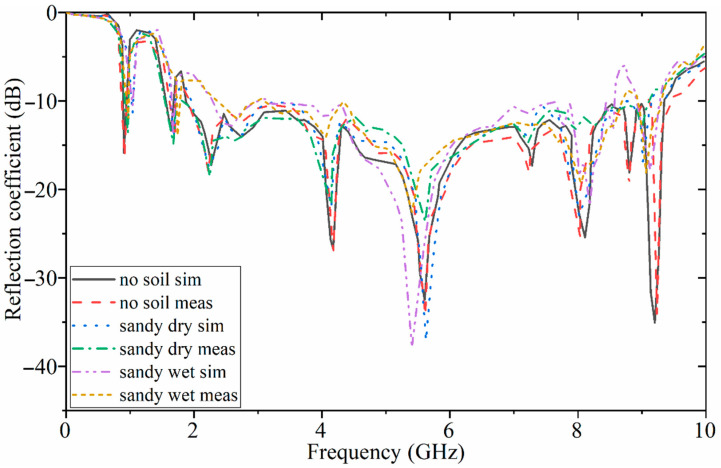
The simulated and measured reflection-coefficient results in terms of different types of soil.

**Figure 11 sensors-22-05183-f011:**
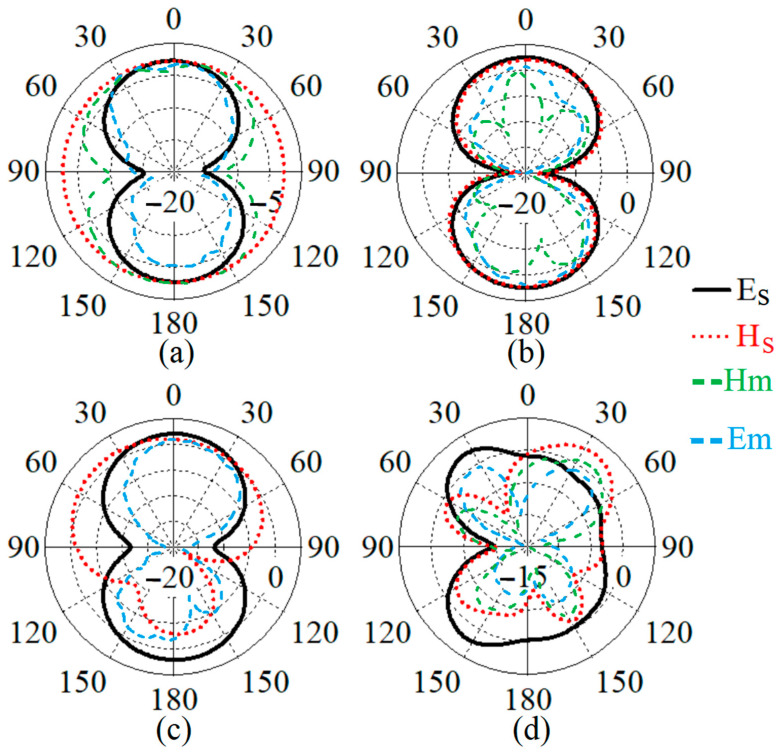
Simulated and measured electric and magnetic fields of the proposed antenna at (**a**) 0.9 GHz, (**b**) 1.8 GHz, (**c**) 1.9 GHz, and (**d**) 9.2 GHz (Es and Hs are the simulation results, and Em and Hm are the measurement results).

**Figure 12 sensors-22-05183-f012:**
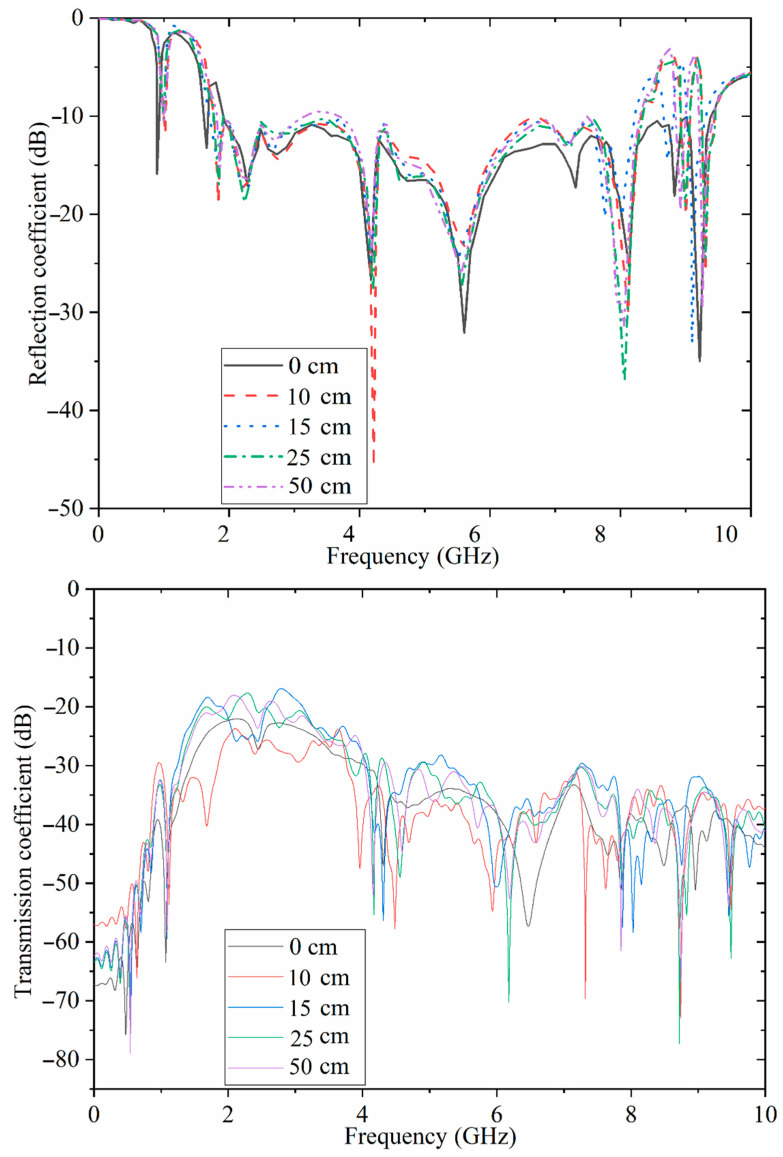
The simulated S-parameters result in the antenna being at a different distance from the soil.

**Figure 13 sensors-22-05183-f013:**
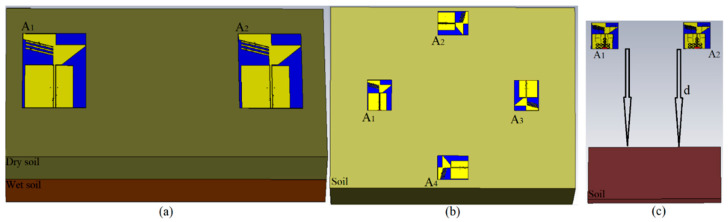
Simulation setup for (**a**) different layers of soil, (**b**) different types of soil with four arrays of antenna, and (**c**) different distances between antennas and soil, (**d**) is the space between the arrays and the soil.

**Figure 14 sensors-22-05183-f014:**
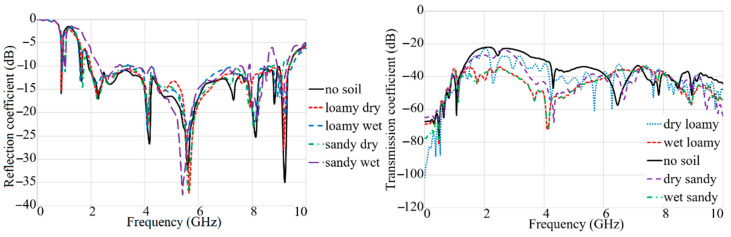
The simulated result of reflection and transmission coefficient results of the proposed antenna in different soils.

**Figure 15 sensors-22-05183-f015:**
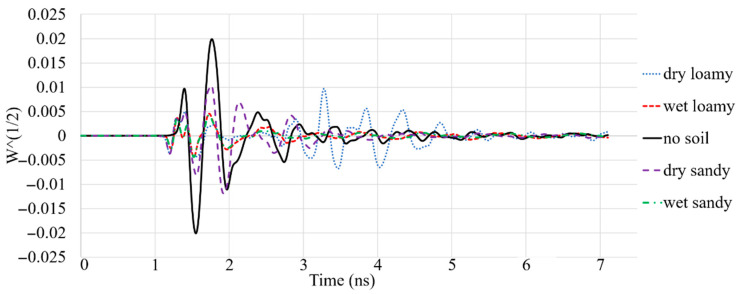
The simulated received signals when passed through different types of soil.

**Figure 16 sensors-22-05183-f016:**
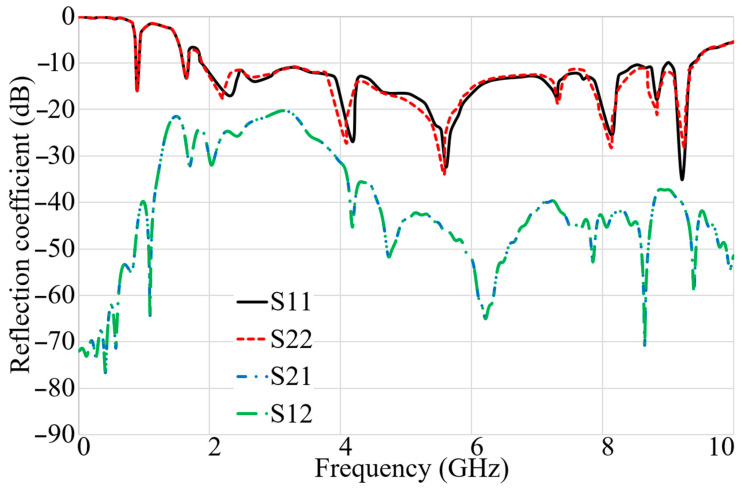
The simulated S-parameter results (reflection coefficients: S_11_, S_22_; transmission coefficients: S_21_, S_12_) when two layers of wet and dry soil exist.

**Figure 17 sensors-22-05183-f017:**
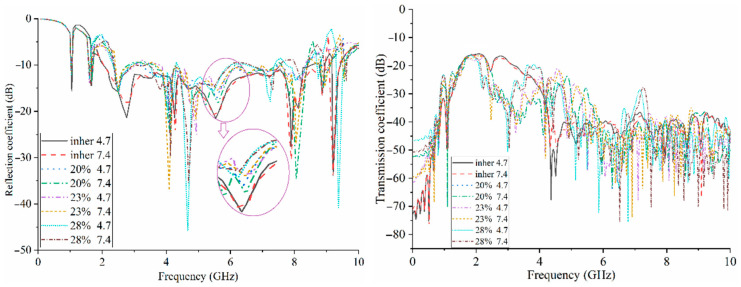
The simulated S-parameter results for different moisture contents.

**Figure 18 sensors-22-05183-f018:**
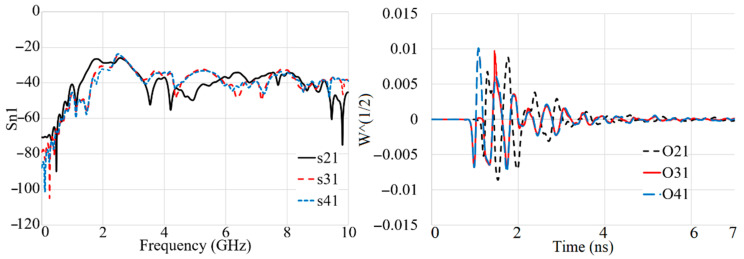
The simulated transmission coefficient (S_21_–S_41_) results and received signals (O_21_–O_41_) when three targets exist.

**Figure 19 sensors-22-05183-f019:**
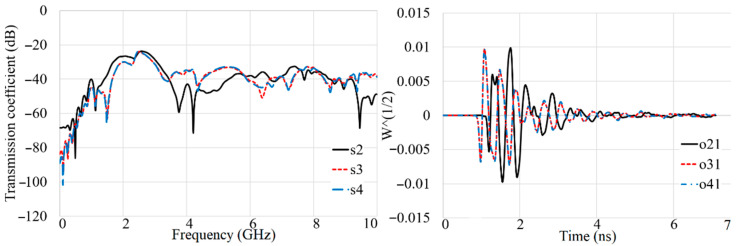
The simulated transmission coefficient (S_21_–S_41_) results and received signals (O_21_–O_41_) when no target exists.

**Figure 20 sensors-22-05183-f020:**
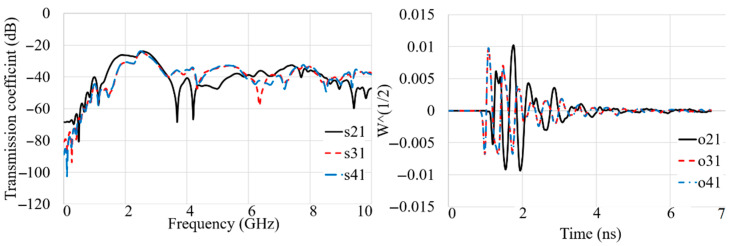
The simulated transmission coefficient (S_21_–S_41_) results and received signals (O_21_–O_41_) when an off-center target exists.

**Figure 21 sensors-22-05183-f021:**
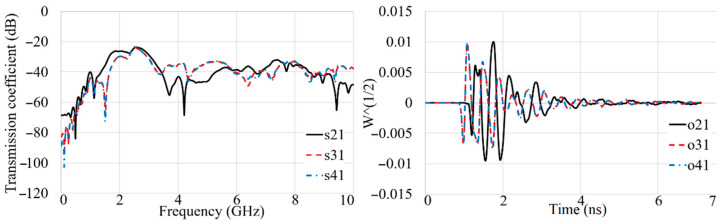
The simulated transmission coefficient (S_21_–S_41_) results and received signals (O_21_–O_41_) when a central target exists.

**Figure 22 sensors-22-05183-f022:**
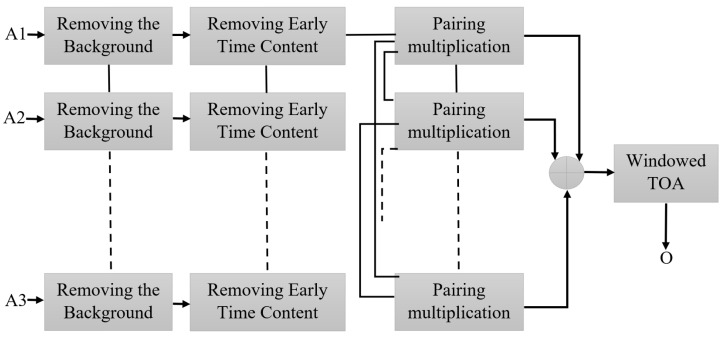
A robust time-reversal algorithm [45].

**Figure 23 sensors-22-05183-f023:**
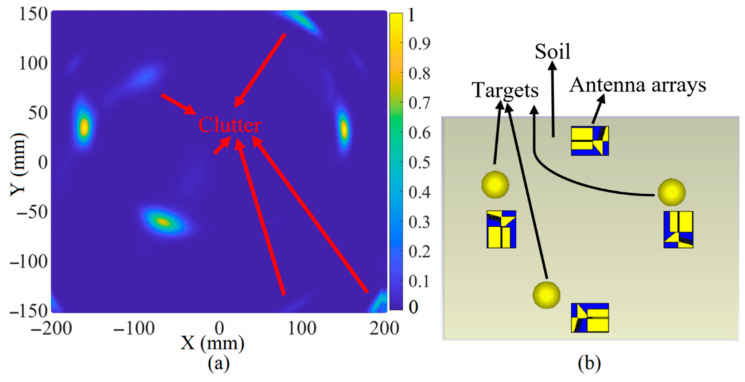
The reconstruction image of the target when three targets exist using simulated received signals (**a**) is the reconstructed image and (**b**) is the simulation setup.

**Figure 24 sensors-22-05183-f024:**
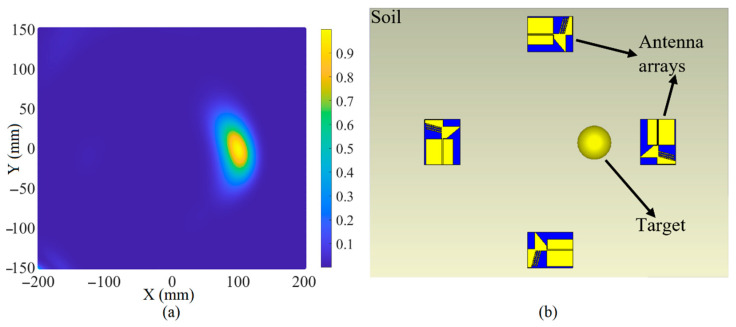
The reconstruction image of the target when an off-center target exists using simulated received signals (**a**) is the reconstructed image and (**b**) is the simulation setup.

**Figure 25 sensors-22-05183-f025:**
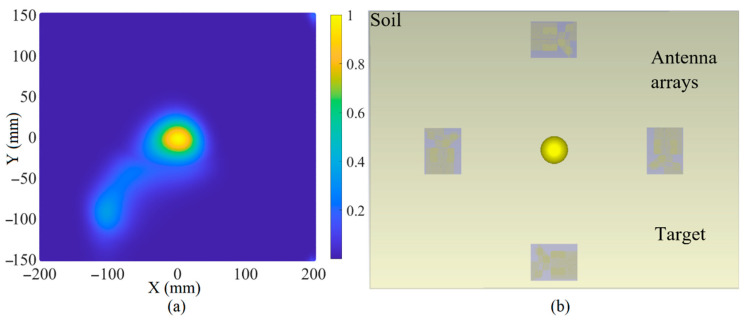
The reconstruction image of a central target using simulated received signals (**a**) is the reconstructed image and (**b**) is the simulation setup.

**Figure 26 sensors-22-05183-f026:**
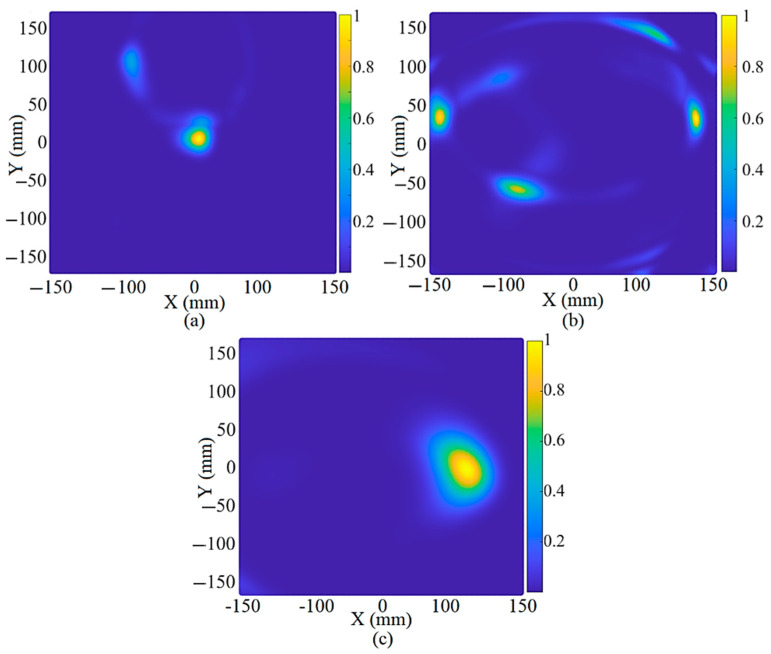
The reconstruction image of (**a**) central target, (**b**) three targets, and (**c**) off-center target using the measured signals.

**Table 1 sensors-22-05183-t001:** The optimized dimensions of the proposed antenna.

Parameter	Values (mm)	Parameter	Values (mm)
$L_{s}$	50.00	$W_{p2}$	10.00
$L_{f}$	28.80	$W_{p3}$	5.40
$L_{g1}$	28.25	$S_{1}$	16.00
$L_{g2}$	28.55	$S_{2}$	8.40
$W_{g1}$	17.75	$S_{3}$	10.00
$W_{g2}$	10.75	$S_{4}$	5.30
$W_{p1}$	4.70	$S_{5}$	4.70
$W_{s}$	39.5	$r_{2}$	1.5
$r_{1}$	2.5	$L_{1}$	8.00
$L_{2}$	8.00	$L_{3}$	19.50
$L_{4}$	17.85	$L_{5}$	6.70
$L_{6}$	19.00	$L_{7}$	7.5
$L_{8}$	3.85	$L_{9}$	1.00
$W_{f}$	1.5		

**Table 2 sensors-22-05183-t002:** The proposed antenna’s performance compared with similar works for the same application.

Ref.	BW (GHz)	Dimensions (mm^2^)	Gain (dBi)	Applications
[9]	1.75–10	120 × 130	17 (at 1.1 GHz)	UWB GPR
[12]	0.98–4.5	107.7 × 68	10.3 (at 3 GHz)	UWB GPR
[34]	1.57–7.04	90 × 100	N/A	UWB-GPR
[35]	1.7–475	200 × 100	8.39 (at 3 GHz)	UWB, GND, and through-wall imaging
[36]	0.5–4	173 × 299	7.46 (at 1 GHz)	UWB GPR
[37]	0.64–1.97	177.75 × 91.5	5.1 (at 1.6 GHz)	UWB GPR
[38]	1.8–2.6	50 × 80	N/A	UWB GPR
[39]	0.45–10	240 × 240	6.68 (at 0.8 GHz)	UWB GPR
[40]	0.42–5.5	172 × 230	7.96 (at 4.8 GHz)	UWB GPR
This work	0.9, 1.8, 1.9–9.2	50 × 39.5	10.8 (at 3 GHz)	UWB GPR

**Table 3 sensors-22-05183-t003:** The proposed antenna’s performance comparison table with some similar works that used soil as medium and test bed.

Ref.	Dimensions (mm^2^)	Max Gain (dBi)	BW (GHz)	Examined Material and Thickness (cm)
[9]	130 × 120	17	0.25–10	Soil (100)
[12]	107.7 × 68	10.3	0.98–4.5	Soil, 12
[41]	180 × 220 × 50	7	0.6–4.6	Sandy soil (N/A)
[42]	235 × 270	<8	0.18–6.2	-
[43]	78.5 × 47.9	<9	3.2	Asphalt, base, sub-grade (max 30 cm)
Proposed antenna	50 × 39.5	10.8	0.9, 1.8, 1.9–9.2	Loamy soil (wet and dry), sandy soil (wet and dry) (max 50)

## Data Availability

The codes and extracted data are available under the permission of the authors and associated universities.

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
