# Peer review of "High Gain Compact UWB Antenna for Ground Penetrating Radar Detection and Soil Inspection"

_sensors, 2022, doi:10.3390/s22145183_

Round 1
Reviewer 1 Report
The main characteristic of the proposed antenna is claimed to be its high gain and compact size. However, it is not described and compared enough to the most prominent works in this field.
Table 1 should include prominent similar papers for comparison, especially it should be shown at which frequency the antenna gains are reported.
I don't believe the proposed antenna is compact enough compared to the previous UWB structures which have been published in abundance before.
Besides, the Authors should clarify how the high gain of an antenna can improve the image quality of GPR radar. Is it essential?
There are a lot of English language mistakes (punctuation, grammatical, etc). It needs a great amount of editing.
Author Response
Thank you very much for you comments.
Reviewer 1:
The main characteristic of the proposed antenna is claimed to be its high gain and compact size. However, it is not described and compared enough to the most prominent works in this field.
Response: Another table of some related works is presented based on their applications, gain, and size. Table 3 is the comparison among those that have been used for UWB GPR application and the used different kinds of soil with different thicknesses. Table 2 compares the proposed antenna with several antennas designed for UWB GPR applications only.
Corrected: page 12. Lines: 364-370
Tables 2 and 3 show the comparison Tables. Table 2 demonstrates the comparison performances between several similar works. All these works designed a UWB antenna for GPR applications. They considered both low and high-frequency bands. The proposed antenna showed better BW and gain performance with lower dimensions. On the other hand, Table 3 compares our antenna with some antennas designed for GPR application, and they were considered in a medium like soil with different thicknesses. The proposed antenna offered better performance in comparison with them too.
Table 2. The proposed antenna’s performance compared with similar works for the same application
Ref |
BW (GHz) |
Dimensions (mm2) |
Gain (dBi) |
Applications |
[9] |
1.75-10 |
120*130 |
17 (at 1.1 GHz) |
UWB GPR |
[12] |
.98-4.5 |
107.7*68 |
10.3 (at 3 GHz) |
UWB GPR |
[34] |
1.57-7.04 |
90*100 |
N/A |
UWB-GPR |
[35] |
1.7-475 |
200*100 |
8.39 (at 3 GHz) |
UWB, GND, and through-wall imaging |
[36] |
.5-4 |
173*299 |
7.46 (at 1 GHz) |
UWB GPR |
[37] |
.64-1.97 |
177.75*91.5 |
5.1 (at 1.6 GHz) |
UWB GPR |
[38] |
1.8-2.6 |
50*80 |
N/A |
UWB GPR |
[39] |
.45-10 |
240*240 |
6.68 (at 0.8 GHz) |
UWB GPR |
[40] |
.42-5.5 |
172*230 |
7.96 (at 4.8 GHz) |
UWB GPR |
This work |
0.9, 1.8, 1.9 – 9.2 |
50 × 39.5 |
10.8 (at 3 GHz) |
UWB GPR |
Table 1 should include prominent similar papers for comparison, and especially it should be shown at which frequency the antenna gains are reported.
Response: As requested, another table was added to show a better comparison between our work and some similar articles.
Corrected: page 12,13. line 384-385
Table 2. The proposed antenna’s performance compared with similar works for the same application
Ref |
BW (GHz) |
Dimensions (mm2) |
Gain (dBi) |
Applications |
[9] |
1.75-10 |
120*130 |
17 (at 1.1 GHz) |
UWB GPR |
[12] |
.98-4.5 |
107.7*68 |
10.3 (at 3 GHz) |
UWB GPR |
[34] |
1.57-7.04 |
90*100 |
N/A |
UWB-GPR |
[35] |
1.7-475 |
200*100 |
8.39 (at 3 GHz) |
UWB, GND, and through-wall imaging |
[36] |
.5-4 |
173*299 |
7.46 (at 1 GHz) |
UWB GPR |
[37] |
.64-1.97 |
177.75*91.5 |
5.1 (at 1.6 GHz) |
UWB GPR |
[38] |
1.8-2.6 |
50*80 |
N/A |
UWB GPR |
[39] |
.45-10 |
240*240 |
6.68 (at 0.8 GHz) |
UWB GPR |
[40] |
.42-5.5 |
172*230 |
7.96 (at 4.8 GHz) |
UWB GPR |
This work |
0.9, 1.8, 1.9 – 9.2 |
50 × 39.5 |
10.8 (at 3 GHz) |
UWB GPR |
I don't believe the proposed antenna is compact enough compared to the previous UWB structures which have been published in abundance before.
Response: The antenna has been compared with those considered for both UWB and GPR applications. It is not designed for UWB communications only since considering a medium like soil is crucial for this work. Therefore, at the time of the comparison, those works were also picked up that design for GPR application.
Besides, the Authors should clarify how the high gain of an antenna can improve the image quality of GPR radar. Is it essential?
Response: An antenna with high gain might not be essential for UWB and GPR applications. But it is an important factor for an antenna because antenna gain is more commonly quoted than directivity in an antenna's specification sheet because it considers the actual losses that occur. In addition, the antenna gain indicates how strong a signal an antenna can send or receive in a specified direction. Gain is calculated by comparing the measured power transmitted or received by the antenna in a specific direction to the power transmitted or received by a hypothetical ideal antenna in the same situation. Antenna gain is also a measure of the maximum effectiveness with which the antenna can radiate the power delivered by the transmitter towards a target. Besides, the signal power is vital in signal processing, and imaging of a target here since the received signals are used to reconstruct the image and are utilized in the algorithm. Due to the gain importance in antenna design, several works considered high gain while designing an antenna for radar and microwave imaging. Some of these works are as follows:
- K. Pandey, H. S. Singh, P. K. Bharti, A. Pandey, and M. K. Meshram, “High Gain Vivaldi Antenna for Radar and Microwave Imaging Applications”, International Journal of Signal Processing Systems Vol. 3, No. 1, June 2015.
Ebenezer Tawiah Ashong, Baidenger Agyekum Twumasi, Patrick Kwashie Kagbetor, “Compact High Gain Elliptical Patch Antenna for Satellite Synthetic Aperture Radar”, 2021 IEEE 8th International Conference on Adaptive Science and Technology (ICAST -2021), Nov. 25 – 26, 2021, Accra, GHANA.
Shams Khaled Ahmed and Zaid A Abdul Hassain, “Gain enhancements of tapered slot Vivaldi antenna for radar imaging applications”, Journal of Physics.
Madan L. Meena, Mithilesh Kumar, “Design of high gain/directional ultra-wideband antenna for radar imaging systems”, Volume29, Issue2, February 2019.
Vanaja Selvaraj, Poonguzhali Srinivasan, Jegadish Kumar.K.J, Rahul Krishnan, Karunakaran Annamalai, “Highly Directional Microstrip Ultra-Wide Band Antenna for Microwave Imaging System”, Acta Graphica Vol 28, No 1 (2017); 35-40.
Artit Rittiplang and Pattarapong Phasukkit, “1-Tx/5-Rx Through-Wall UWB Switched-Antenna-Array Radar for Detecting Stationary Humans”, Sensors (Basel). 2020 Dec; 20(23): 6828.
Corrected: page 12, lines 371-381.
There are a lot of English language mistakes (punctuation, grammatical, etc). It needs a great amount of editing.
Response: the English of the paper was checked with ‘Grammarly Premium’ and then double-checked with a proofreader.
Corrected:

Reviewer 2 Report
This innovative ultra-wide bandwidth antenna with very small size should be interesting to the readers. The results from the testing and experiment are also supportive. Your article was well prepared and so I don’t have the too much constructive comments on your article except of the following:
1. The conclusion section should be further improved with quantitive and concrete description. Now, the part in this version is not a strong conclusion.
2. Check with many figures, like fig. 7/15/17/18/19/20. The description about the acronyms for each curve was missing.
3. The acronym BW should be given the full name when it appears firstly in the article.
Author Response
Thank you very much for you comments.
Reviewer 2:
This innovative ultra-wide bandwidth antenna with a very small size should be interesting to the readers. The results from the testing and experiment are also supportive. Your article was well prepared and so I don’t have the too much constructive comments on your article except of the following:
- The conclusion section should be further improved with quantitative and concrete descriptions. Now, the part in this version is not a strong conclusion.
Response: the conclusion was improved.
Corrected: page 22, lines 622-659
The GPR statement is given to a system that can find a hidden or buried object using the scatters from the electromagnetic waves sent by antennas. Most of these systems have utilized a low-frequency band (for better penetration in soil, for example) and bulky the sending and receiving systems (antennas). Therefore, a low-profile antenna with high performance is required. A novel Ultra-wideband paddle shape antenna incorporated with periodic metamaterial array structures is proposed for landmine detection using Ground Penetrating Radar (GPR) system. The proposed antenna is designed at a center frequency of 5 GHz on a PTFE substrate () beginning with a conventional rectangular patch fed by using the CPW technique. It is to meet the UWB antenna's working band based on the FCC. Then, the ground was cut asymmetrically to reduce the capacitive coupling (enhance BW at a higher frequency and thoroughly broaden the antenna impedance bandwidth). Afterward, the patch is cut, making a paddle shape to reduce the surface wave (another factor degrading the antenna's performance). Eight chamfered-edge cubes then load the antenna following the antenna’s SCD around the paddles, TL, and the GND (it improves the impedance BW at the lower and higher end of the working BW). Next, the antenna was loaded with three stubs (to convert the slight stopbands to passband after adding the cubes, to improve the reflection coefficient level of two more poles at 0.9 GHz and 1.8 GHz created before), and four shorting pins (to reduce the possible surface waves around the stubs and patch). Last but not least, an SRR periodic MTM structure and six more pins were utilized to enhance the radiation efficiency and gain, along with improving the level of the reflection coefficient. As a result, it offers enhanced gain and directivity.
The proposed antenna offers an operational bandwidth from 1.9 to 9.1 GHz. The average directivity reaches 11.2 dBi, while the gain and radiation efficiencies are 10.8 dBi and 97%, respectively. The size is small due to the antenna loading with stubs and even resonating at two more resonances at lower frequency bands of 0.9 GHz and 1.8 GHz working for ISM. The antenna performance was simulated and then compared with the measurement. Good agreement exists between numerical and experimental results. The antenna response is studied for ground coupling GPR applications like finding a hidden metallic target considering various considerations such as different types of soil, different levels of moisture content of the soil, soil with multiple layers, and different thickness of the soil (the space between the antennas and the level soil, target, and beneath the level of the soil in the container). Stable and linear transfer function response and flat group delay response is obtained in the antenna passband, which confirms the low dispersive nature of the proposed UWB antenna and thus ensures its operational capability as a GPR antenna. Moreover, the reconstructed GPR image collected with the proposed antenna from the simulated and experimental setup with a sandbox, metallic target, and the moist sandy soil layer, shows that the proposed antenna is a reliable candidate for GPR applications.
- Check with many figures, like fig. 7/15/17/18/19/20. The description about the acronyms for each curve was missing.
Response: all the acronyms were checked and added in figures and throughout the text.
Corrected: throughout the text.
- The acronym BW should be given the full name when it appears firstly in the article.
Response: It was given in the abstract.
Corrected: page 1.

Reviewer 3 Report
This manuscript presents a UWB antenna for application on ground penetration analysis.
The works seems interesting, but few issues must be clarified before giving the full recommendation for publicating this manuscript on these transactions.
On abstract, what is meant with "and two more resonances at 0.9 and 1.8 GHz is 26
also achieved"? Are other resonances present that are not cited?
On abstract remove the mention to IoT application and it is out the scope of the work that is the antenna sensor.
The measurement setup on Figure 1 is very difficult to understand without a full discussion of its implementation, e.g., the equipments, acessories and so on. A photo explaning the setup is also advisable.
On page 4 it is stated that the used substrate is based on a marketable cheap PTFE substrate. How cheap? And which is the manufacturer and reference of this material?
On section 2.1 describing the parametric study it was interesting to include references to the theory explaining the effect of feeder's length and width.
On page 6 it is stated that "Pin loading creates more poles and slightly increase the BW". Why?
Make a better explanation of the proceedure followed in the simulations of Figure 6, as well as, how the different parameters and meshes were trimmed.
In the beggining of section 3, please explain the selection of these mediuns (appart from the "classic" free space).
Who is the reproducibility of the experiments?
What is the miniminum feature detectable in the several mediuns.
How robust is the sensor antenna to false targets embedded within the soil? These experiments were carried?
At last but not least, It is not clear if in monostatic state the same antenna is used as both the transmitting and receiving antenna. It is also not so clear if both in the bistatic and multistatic states the receiving antennas are the same type of transmitting antenna.
Author Response
Thank you very much for your comments.
Reviewer 3:
This manuscript presents a UWB antenna for application in ground penetration analysis.
The works seems interesting, but few issues must be clarified before giving the full recommendation for publication this manuscript on these transactions.
On abstract, what is meant with "and two more resonances at 0.9 and 1.8 GHz is 26 also achieved"? Are other resonances present that are not cited?
Response: It means that apart from the wide BW obtained from 1.9 GHz to 9.2 GHz, two more resonances at 0.9 GHz and 1.6 GHz was also achieved after loading the antenna. Therefore, the antenna works for two more application bands: ISM and L-band.
Corrected: page 1, line 27.
The antenna is designed on a thin layer of economic Polytetrafluoroethylene (PTFE) substrate with dimensions 50×39×0.508 mm3 and works in the frequency range of 1.9- 9.2 GHz. In addition, two more resonances at 0.9 and 1.8 GHz are also achieved (Hence, the antenna works for more than two application bands as ISM and L-band).
On abstract remove the mention to IoT application and it is out the scope of the work that is the antenna sensor.
Response: The statement was removed.
Corrected: Page 1, line 32.
The measurement setup on Figure 1 is very difficult to understand without a full discussion of its implementation, e.g., the equipment’s, accessories and so on. A photo explaining the setup is also advisable.
Response: The full explanation was added for both S-parameters and radiation pattern measurement. Then, Figure 1 was improved. Finally, Figure 2 was added for a better explanation.
Corrected: page 3, lines 106-121.
Figure 1c is the fabricated antenna. After fabrication, the antenna is measured using VNA shown in Figure 1d. The antenna is measured in the air first, and then the media is changed into the soil. The antenna is connected to the first terminal of the VNA, and another antenna is connected to terminal 2 to carry out the measurement. After securing the antennas to the VNA’s terminals through cables, the VNA should be calibrated using the calibration kits based on the diameter of the antenna's SMA ports (2.4 mm or 3.5 mm). After assigning the frequency band and the calibration, the antenna’s S-parameters (reflection and transmission coefficients) are measured. Another part of the measurement is the measurement of the radiation pattern of the antenna (Figure 2). The proposed antenna is fixed on the rod at the center, and a reference horn antenna (connecting to the power meter) is specified on the rotating rod to perform the radiation pattern measurement. The motion controller, which rotates the rotating rod, does the rotating in each step of 3 degrees like it stops in each stage of 3 degrees and record and then continue. The recorded power by the meter is connected to MATLAB code in the PC, and then the radiation pattern is drawn). Finally, the antenna is moved to the elevation plane, and the measurement process starts again.
On page 4 it is stated that the used substrate is based on a marketable cheap PTFE substrate. How cheap? And which is the manufacturer and reference of this material?
Response: the statement of being cheap was compared to a substrate like Rogers 5880 [23-24].
Corrected: page 4, lines 131-132
The reference was added.
On section 2.1 describing the parametric study it was interesting to include references to the theory explaining the effect of feeder's length and width.
Response: the requested references were added to support the statement.
Corrected: page 6, lines 207-208.
The related reference regarding the theory for the impacts of feed line width and length [25-27].
On page 6 it is stated that "Pin loading creates more poles and slightly increase the BW". Why?
Make a better explanation of the procedure followed in the simulations of Figure 6, as well as, how the different parameters and meshes were trimmed.
Response: utilizing shorting pins is a technique that can be applied for producing more poles or resonances, enhancing the antenna gain and radiation efficiency, increasing the BW, and surface-wave suppression.
The surface current distribution was improved.
Different parameters and amounts of mesh cells were also explained. The mesh cells are chosen in CST studio software enough to have more accuracy in the simulated results. It's more than 1.5 million in total. Having more mesh cells in simulation enhances the chances of having more agreement between the simulated and measured results.
Corrected:
page 7, lines 247-252
Hence, it is loaded with four pins. Pin loading creates more poles (resonances). Therefore, it slightly increases the BW (utilizing shorting pins is a technique that can produce more poles or resonances, enhancing the antenna gain and radiation efficiency, increasing the BW and surface-wave suppression). In addition, due to the inductive shunt effect of these shorting pins, the dominant mode's resonant frequency is tuned up to enhance the radiation gain of a single patch antenna [28-30].
Page 8, lines: 271-290
It can be seen that the magnetic field strength is low as the light green color indicates 0 to 3 A/m. The current distribution has a shallow flow and magnitude at the input port, and the current distribution reduces as it approaches the patch. The signal from the input port is partially lost as it travels to the patch. The SCD for the chamfered cubes also shows strong density around these cubes at 2.1 GHz and 9 GHz, where both ends of the BW exceeded what was obtained for the previous design stage. Most likely other parts of Figure 7 depict the strong SCD at the poles in the working BW at each design stage. For instance, the surface current density around the stubs at 0.9 GHz is higher than the current density at 1.6 GHz. It can be deduced that the electromagnetic wave of a specific frequency is excited by the stubs and the patch resulting in resonance at 0.9 GHz as the color is nearly red and the magnetic field around the stubs at that frequency is increased to 8 A/m. Figure 7d shows the surface current distribution around the MTM structure at 1.9 GHz and 9.2 GHz. It can be observed that the current is no longer limited between the radiation patch and feedlines. It also distributes around SRR, demonstrating that the SRR can act as a resonator to generate new resonance at 1.9 GHz. Obviously, after adding the SRR, the current distribution is changed.
Moreover, the current shocks back and forth in the SRR, radiating a specific electromagnetic wave frequency. At the frequency of 1.9 GHz, the surface current mainly distributes around the SRR. Thus, the newly generated resonance is primarily affected by the parameters of the SRR.
Page 7, lines: 227-235
The other parameters of the antenna design were also optimized to achieve the best results (The optimization is performed in CST using the PSO algorithm after the actual values obtained for each parameter). For instance, the length of the stubs (L4) should not exceed 17.85 mm (increase the undesired capacitive coupling) and less than 4 mm (affects the reflection coefficient level of the lower bands like 0.9 GHz and 1.8 GHz). The width of the stubs (L9) affects the impedance bandwidth of the antenna at the lower end of the working BW and two lower resonances. The space between each stub (S5) also needs some attention as it should not be more than 4.7 mm and less than 4. Less than 4 mm increases the surface waves around that area.
In the beginning of section 3, please explain the selection of these mediums (apart from the "classic" free space).
Response: the selection explanation was added.
Corrected: page 10, lines: 323-329.
These soils were sandy and loamy for wet and dry conditions. They were bought from the shop to fill the container with different soil and various moisture content levels. The moisture content of the soil is measured first. Then, it's calculated by weighing the wet soil sampled from the field, drying it in an oven, and then weighing the dry soil. Thus, gravimetric water content equals the wet soil mass minus the dry soil mass divided by the dry soil mass. In other words, the mass of the water is divided by the mass of the soil.
Who is the reproducibility of the experiments?
Response: The reproducibility is opened to anyone to replicate/repeat the study and get the same results as are expected each time the experiment/study is done because it was verified by simulation and measurements as details entailed. It is reproducible by any whom repeats the study.
Corrected:
What is the minimum feature detectable in the several mediums?
Response: The target considered and detected here is 25 mm diameter. The minimum target that the antenna can detect, is calculated by resolution parameters like range, resolution, cross resolution, and spatial resolution.
Corrected: page 20, 21, lines 615-620
In addition, the smallest target that the antenna can detect in any location within a certain distance is around 10 mm in diameter. It can be calculated considering the spatial resolution. Apart from the spatial resolution, range resolution (10 mm for 10 cm distance between the antenna and the target) and cross-range resolution (11.28 mm). The full explanation and formulas are presented in [30, 52-54].
How robust is the sensor antenna to false targets embedded within the soil? These experiments were carried?
Response: The robustness of the antenna on a false target can be shown by how much the antenna can be accurate on the detection of the actual target and some clutter. Can the target be differentiated from the other objects in the testing medium, like when we have three targets, for example? It can be concluded that the antenna can detect false targets and differentiate them from other targets or objects. Figure 26 proved that. The clutters that were detected in Figure 26a are due to the uneven distribution of the soil in the test bed. This uneven distribution can be like a situation when another particle exists in the soil. Another proof of the robustness of the antenna is shown in Figure 26b, when all three targets were detected. However, some clutters were also detected. But the targets are perfectly recognizable since their dynamic range is nearly one (yellow color).
Corrected: page 21, lines 602-612
At last, but not least, It is not clear if in monostatic state the same antenna is used as both the transmitting and receiving antenna. It is also not so clear if both in the bistatic and multistate states the receiving antennas are the same type of transmitting antenna.
Response: the transmitter and receivers send and receive the signals in a multistatic manner, and all the antennas for the receiver and transmitter are identical.
Corrected: page 17, lines 479-484
It should be mentioned that the transmitters and receivers send and receive the signals in a multistatic manner and all the antennas as receivers and transmitters are identical. One transmitter sends, and the other receives, and then the transmitter switches until the 4th array (as a usual multistatic). However, to reconstruct the image of the target in this article, only one transmitter sends, and the others receive, thus saving processing time.

Round 2
Reviewer 1 Report
A comprehensive edit has been done and the reviewer remarks have been answered and suitable edits were added to the paper. The paper can now be published in its current form.